# Surgical Management of a Salter-Harris Type I Distal Physeal Fracture of the Tibia in a Calf: A Case Report

**DOI:** 10.3390/vetsci10070463

**Published:** 2023-07-14

**Authors:** Victor Lemaitre, Emeline Cote, Christophe Bellon, Hervé Cassard, François Schelcher, Renaud Maillard, Rodolphe Robcis

**Affiliations:** 1Clinic of Ruminants, Ecole Nationale Vétérinaire de Toulouse, 31076 Toulouse, France; emeline.cote@live.fr (E.C.); renaud.maillard@envt.fr (R.M.); rodolphe.robcis@envt.fr (R.R.); 2Veterinary Clinic of Decize, 58300 Decize, France; clinique.champvert@orange.fr

**Keywords:** orthopedics, physeal fracture, tibia, Salter-Harris type I, external fixators, surgery, calf, case report

## Abstract

**Simple Summary:**

Fractures are common conditions in cattle, especially in calves. However, a few reports about physeal fractures are available in the literature. The aim of the study is to provide accessible and successful care of a distal physeal fracture of the tibia by using external fixators.

**Abstract:**

Fractures are common conditions in cattle, including tibial fractures. Physeal tibial fractures are more specific and less frequently met in field conditions. A calf with a Salter-Harris type I distal physeal fracture of the tibia was referred to the National Veterinary School of Toulouse (ENVT), France. Although the use of external fixators in the treatment of tibial fractures is common, distal physeal tibial fractures require a different and specific technique involving them. They were first used as a lever arm to reduce the fracture due to the severe displacement. A hock joint bypass was then performed. Six weeks after treatment, the calf recovered successfully from the use of the affected limb without any adverse sequelae. The present case provides management of a distal tibial fracture using external fixators. This innovative and accessible surgical technique may be used by veterinary practitioners in future similar cases of distal tibial fractures when pins in the distal end cannot be inserted.

## 1. Introduction

Lameness in cattle is a common clinical entity [1]; with calves, the most common causes are associated with bone fractures and arthritis [2]. Fractures can vary widely regarding location, type and nature; those associated with the tibia are, perhaps, the most frequently encountered.

Epiphyseal fractures are common in calves, especially those older than six months [3]. Most of the previous cases of physeal fractures described in calves are located on the distal physis of the metacarpus and metatarsus [3] and on the proximal capital physis of the femur [4,5,6]. According to the Salter-Harris nomenclature, types 1 and 2 are the most common. To the best of the authors’ knowledge, a very small number of similar fractures have been described which suggests the case reported in this paper is rare [7].

The present paper describes the surgical management of a Salter-Harris type 1 distal physeal fracture of the tibia in a pedigree heifer calf using external fixators.

## 2. Case Presentation

A 35-day-old female Blonde d’Aquitaine calf was referred to the Clinic of Ruminants of the National Veterinary School of Toulouse, France, following the sudden onset of lameness involving the left hind limb two days previously. The farmer had not observed any prior trauma to the heifer and throughout its illness had suckled its dam with vigor. On arrival at the clinic, the calf weighed 80 kg (cattle scale, Clinic of Ruminants, Toulouse), which is in accordance with her age. A general clinical examination revealed no abnormal findings other than being unable to support the affected left hind limb: when asked to move, a severe lameness was observed. However, its behavior appeared relatively unaffected as it tried to walk, and could stand, lie down and get up, all unaided. The left hock presented as swollen with a hematoma over the distal tibia and there was an abnormal angle inwards towards the medial aspect of the limb. There was no palpable bone splintering nor break in the skin over the joint.

A provisional diagnosis included a fracture involving the distal region of the tibia (diaphysis or epiphysis) and/or the hock joint itself, or either an inter-tarsal or a tibio-tarsal dislocation. An X-ray examination was therefore considered the gold standard ancillary examination for such suspicions.

A definitive diagnosis was made following routine cranio-caudal and latero-medial radiographs that revealed a Salter-Harris type I distal physeal fracture of the tibia (Figure 1). In fact, the growth plate was no longer visible on the distal tibia. The fracture was associated with significant medio-caudal displacement of the proximal end of the tibia.

The severe medial displacement of the tarsal-tibio tarsal joint and the inability to reposition it by traction ruled out the possibility of treating this fracture conservatively; hence, a surgical approach was deemed necessary to restore permanent good function to the limb of this calf—having a high genetic value—with the agreement of its owner. 

Prior to treatment, prophylactic benzylpenicillin–dihydrostreptomycin (intramicine: CEVA) was administered routinely at a dose of 20 mg/kg of body weight (BW), followed by 0.5 mg/kg BW meloxicam (Recocam: Bimeda). General anesthesia was induced using 0.025 mg/kg BW xylazine (Sedaxylan: Dechra) intravenously, 0.06 mg/kg BW butorphanol (Torbugesic: Zoetis) and 2.9 mg/kg BW ketamine (Ketamine 1000: Virbac) [8]. A 12 mm diameter endo-tracheal tube was placed routinely and anesthesia was maintained using isoflurane (IsoFlo: Zoetis) and oxygen. To further relax the left hind limb, procaine (Procamidor: Axience) was administered in the epidural space (L6-S1 interspace, 8 mL). Throughout the procedure, 0.9% isotonic saline was administered through an intravenous drip at a rate of 10 mL/kg/h.

The entire left hind limb from the stifle to the hoof was prepared routinely for surgery, and the calf was placed in right latero-dorsal recumbency. The objective of the surgical procedure was to use the JAM external fixator ([9], pp. 237–242); [10] to create an assembly known as a transfixing frame, in order to create a hock joint bypass ([9], pp. 288–294); [11]. JAM (for Jean-Alphonse Meynard) is a special technique and material for conventional external fixator applications, including Kirschner pins, union bars and original coaptors composed of two circular flanges. A screw locks the two flanges together, and each side comprises a cylindrical hemi-tunnel with a diameter corresponding to that of the pins used.

The physeal fracture made it impossible to create an assembly involving only the tibia, as no pin can be placed in the distal end (i.e., the epiphysis).

Next, 6 mm stainless steel Kirschner pins were placed into the lateral aspect of the distal tibia, the proximal metatarsal, and the calcaneus (see Figure 2 and Table 1).

A 1 cm skin incision was made with a cold blade #22 on the lateral aspect of the limb, on the future location of the pins.

Three 6-mm Kirschner pins were inserted into the tibia latero-medially. Predrilling for each pin with a 4 mm pin was performed in order to ease the insertion. The first pin was placed approximately at the middle third of the bone. The second one was placed close to the fracture site. The third one was intermediate. These three pins were not in the same plane and form a triangle (Figure 2). In addition, the pins were angled so that they were not all parallel, thereby increasing the rigidity of the assembly and preventing translation. When each of the pins was inserted, the insertion of the two cortical bones must be felt separately. This application was done under irrigation with a saline solution to prevent the heating of bones and surrounding tissues ([9], pp. 288–294); ([9], pp. 272–281); [10].

Then three other pins were similarly inserted into the metatarsal bone. The first pin was placed approximately in the proximal third, the second wire was placed as close as possible to the tarso-metatarsal joint, taking care not to be in an intra-articular position. The third one was intermediate ([9], pp. 272–281); [10] (Figure 2). 

The last pin of interest was inserted latero-medially into the tip of the calcaneus, 2–3 cm from its end.

The pins acted: first, as a lever point to reduce the fracture ([9], pp. 272–281); second, as fixators for the external scaffold to immobilize the joint.

As the fracture was old (two days, see above), persistent contraction of the gastrocnemius muscle and others of the upper thigh made the precise reduction of the fracture very difficult. A 7 cm long medial incision opposite the distal end of the proximal end was made to provide leverage via a Hohmann spreader.

As the reduction was not complete, a final cranio-lateral incision was made to get access to the site of fracture. The exact place of the incision may be seen in Figure 3c. A Hohmann spreader was placed in the fracture, between both ends and was used as a lever by one surgeon to reduce the displacement. Concomitantly, repeated flexion and extension movements were performed by other surgeons to ease the procedure by inducing muscle relaxation and reducing the gap between the diaphysis and the epiphysis of the tibia.

Once the fracture has been reduced and held in place, the two incisions described above were sutured in two different planes (muscular and subcutaneous together, then cutaneous).

Once the tibial-tibia tarsal joint had been replaced in its anatomically correct position, the 6 mm union bars and 24 mm coaptors were placed. The union bars were attached to the pins via the coaptors, forming a triangular frame on both the lateral and medial sides. In addition, small bars connecting the 2 long sides of the triangle were added to improve the rigidity of the assembly. This tibio-metatarsal bypass prevents the animal from bending the leg and therefore movement of the hock, thereby promoting bone healing. Thus, on the lateral side, 5 union bars and 12 coaptors were placed, and on the medial side, 5 union bars and 11 coaptors were placed. In addition, the coaptors must always be oriented towards the outside (in relation to the union bars) and the inside (in relation to the pins) of the assembly. The coaptors and union bars were applied about 1.5 to 2 cm from the skin. Post-operative routine radiological examination confirmed that this joint had been repositioned correctly (see Figure 3a,b).

The protruding bars and pins were cut with a bolt cutter. The whole assembly was disinfected with chlorhexidine (HydeaChlorex^®^ Solution, Savetis, France), focusing on the areas where the pins penetrated the skin, and dried carefully. Finally, the skin over the entire hock joint was bandaged below the scaffold to provide a clean environment for pin insertion and protected the mount when the animal moved and lies down.

Post-operative treatment included intravenous 3000 UI/kg BW gentamicin (G4; Virbac) for three days after surgery, and continuance of benzylpenicillin–dihydrostreptomycin for 20 days. The superficial dressings were changed routinely on days two, seven and nineteen after surgery. Recovery was uneventful, confirmed by radiographs taken one, three and five weeks after the surgery. 

Six weeks after surgery, the heifer was sedated using xylazine as before, the scaffold was dismantled and pins removed, and radiographs were taken (Figure 4). The combination of a good alignment of the two bone portions and the absence of significant periostitis or lysis of the bone cortex attested to the success of the procedure.

The heifer was returned home 69 days following initial treatment. Atrophy of the muscles of the left pelvic limb was discernible, and she walked with a persistent limp: she was closely confined in a well-bedded loose box and physiotherapy was initiated on the farm. Four months after surgery, the physical appearance of the heifer appeared normal and the limp had disappeared (Figure 5).

## 3. Discussion

This paper describes the successful management of a complete physeal fracture regarding the distal physis of the tibia. In cattle, the most frequent cause of such fractures is excessive and misdirected traction applied to overcome fetal posterior presentation and relative oversize in dystocia cases [12,13,14]. In comparison, tibial physeal fractures occur in around 10% of cases in foals [15], most frequently as a result of external traumatic events [16]. In addition, such severe displacement of both bone segments on either side of the fracture has not been described before, making the present case report unique.

The severe displacement of the two bone segments on either side of the fracture, combined with the need for a solid assembly to take account of the vertical and horizontal forces applied to the pelvic limb in cattle as they walk, justified the complex assembly described in this paper. The three most important elements in this assembly, which guarantee its solidity, are (a) the location of the pins, (b) the distance between all of them and (c) the configuration of the triangle on either side of the fracture for the scaffolding to be secure. The first pins inserted should be placed as far as possible from the site of fracture and should be inserted into the diaphysis and cross both cortical bones, at the most central part of the medulla. The second pins should be placed as close as possible from the fracture and they should not be inserted in intra-articular position regarding the pins inserted in the metatarsus. The bone should be penetrated as rapidly as possible, with the lowest possible speed of rotation to prevent bone heating. The value of the angles between each pin is not crucial, as it can vary from one animal to another, depending on its conformation and the specific characterization of the fracture. The coaptors and union bars should be about 1.5 to 2 cm from the skin. If they are too close to the skin, abrasive lesions and ulceration may occur [10].

The success of the treatment depends also on the management of the post-operative period. This calf was kept in a clean box with a severe restriction to its movements that encouraged healing. Both frequent new bandage applications and physiotherapy after pin removal contributed to the positive outcome of this case. Physeal fractures usually heal within four weeks [17], whereas the recovery period, in this case, lasted six weeks; in comparison, foals recover from similar conditions within 6 to 8 weeks [18]. As the displacement of the two bone segments on either side of the fracture was severe, an additional two-week extension was given to ensure proper fracture healing before the removal of the external scaffold. 

Post-operative complications must be considered when discussing the prognosis in such cases. In all species, the major complication is joint stiffness, associated with lameness in some cases, especially when pins are applied for more than four weeks [19]. Full recovery is usually quite good after several months. Paraesthesia of the tibial or dorsal cutaneous nerve may be encountered in around 25% of open dislocations found in humans together with arthritis and degenerative osteoarthritis, in 25% of cases of open dislocations [20]. In this case, the use of fixators could have led to osteomyelitis but maintaining optimum sterility and cleanliness both during surgery and post-operatively ensured a successful treatment outcome. Antibiotic prophylaxis is recommended for all surgery involving the placement of implants [21]. First-generation cephalosporins, cefazoline and cefalexine are the antibiotics of choice [22,23], but their use is prohibited in cattle in Europe, so there are no maximum residue limit values for these two molecules. The antibiotics used in the present case are applicable worldwide. Whittem et al. (1999) [24] showed no difference in efficacy between the use of penicillin and cephalosporin in orthopedic surgery in dogs. Moreover, gentamicin diffusion into bones following intravenous administration is good to excellent [25,26]. The combination of penicillin and gentamicin broadened the spectrum of action and targeted the main germs involved in orthopedic surgery, including *Staphycoccus* spp., *Streptococcus* spp. and enterobacteria [27,28,29]. Gentamicin was used for only 3 days due to the risk of nephrotoxicity [30].

The cost-effectiveness of such an intervention must be considered. This treatment was agreed upon with the farmer who accepted the financial cost, which was substantial. Adaptations are possible to reduce the price. For example, the animal remained in the hospital throughout its convalescence but returning home was placed in a calving pen. This option should be favored in field conditions, by educating the farmer about post-operative procedures associated and regular checks by the veterinarian during the recovery period.

In the present situation, the positive outcome made the care clearly profitable. In addition, profitability is enhanced by the high genetic value of the heifer, which comes from an excellent pedigree in terms of weight development and calving easiness.

## 4. Conclusions

To our knowledge, this is the first report of a Salter-Harris type I physeal fracture involving the distal part of the tibia with a severe displacement of bones. The outcome was positive for the present calf, with the application of osteosynthesis material that served first as a lever to reduce displacement and then as a solid framework to maximize the chances of healing. This management was innovative and never described before. As a result, it may be an additional solution to the resolution of future similar cases met by veterinary practitioners.

## Figures and Tables

**Figure 1 vetsci-10-00463-f001:**
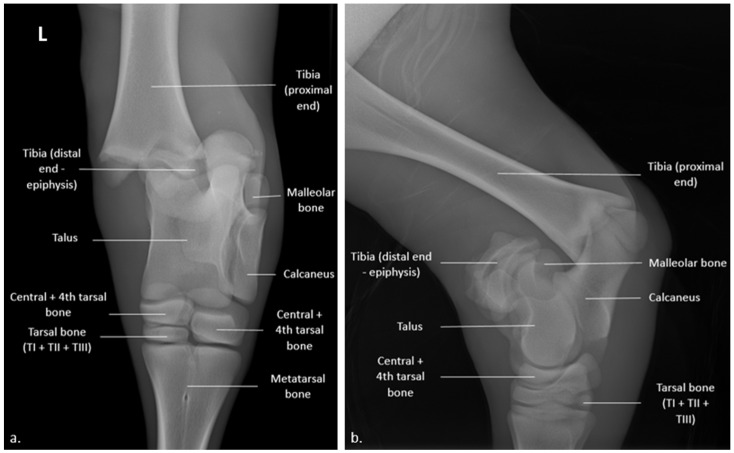
Radiograph of the left hock joint of a 35-day-old heifer calf (74 kV; 6.4 mAs): (**a**) cranio-caudal view with medial displacement of the tibia; (**b**) latero-medial view with posterio-medial displacement, both relative to the talus. The letter “L” in the top left corner indicates that the radiographs regarded the left hind limb.

**Figure 2 vetsci-10-00463-f002:**
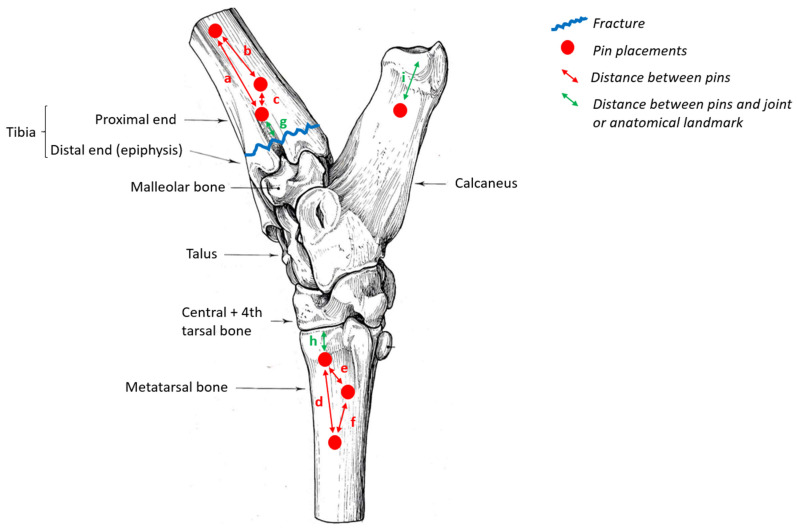
Schematic lateral view of pin locations for the supporting external scaffold to attach to.

**Figure 3 vetsci-10-00463-f003:**
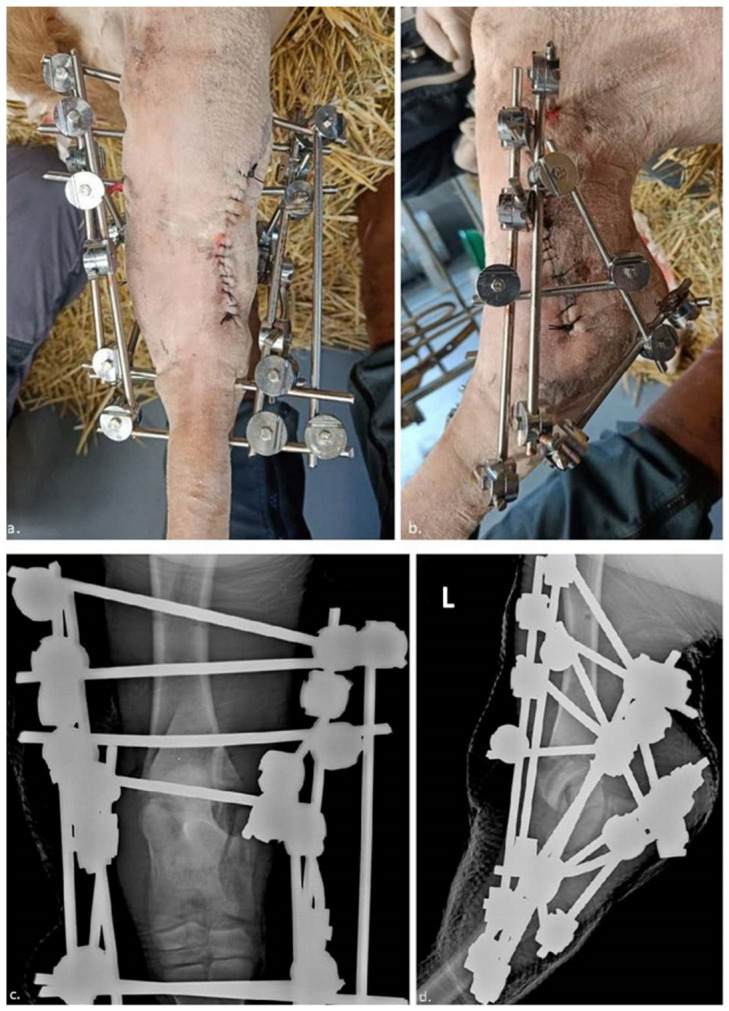
(**a**,**b**): pins and scaffold in place showing the angles between the pins; (**c**) Cranio-caudal and (**d**) latero-medial radiographs describing the pins and external scaffold. The letter “L” in the top left corner indicates that the radiographs regarded the left hind limb.

**Figure 4 vetsci-10-00463-f004:**
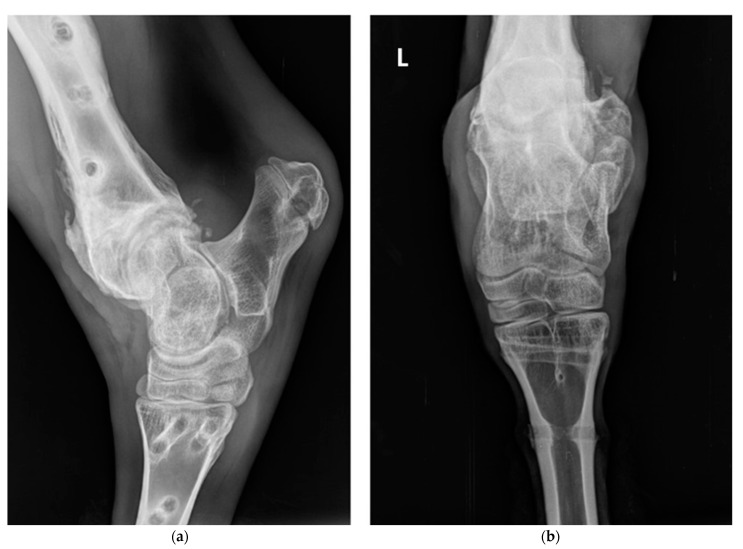
X-rays of the hock after removal of the fixators (74 kV; 6.4 mAs): (**a**) latero-medial and (**b**) cranio-caudal radiographs showing conformation of the tibio-tarsal-metatarsal joint following removal of the pins and scaffold and the absence of significant periostitis or lysis of bone cortex. The letter “L” in the top left corner indicates that the radiographs regarded the left hind limb.

**Figure 5 vetsci-10-00463-f005:**
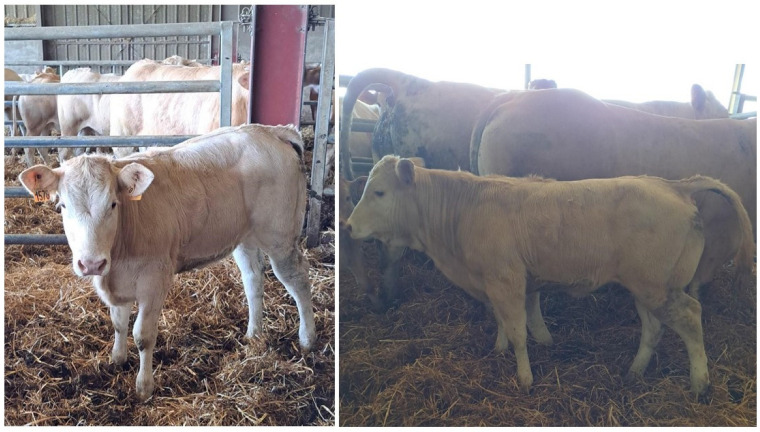
Five-month-old Blonde d’Aquitaine heifer following surgical treatment of a physeal fracture of the tibia. There remains a slight left hind limb extension at rest.

**Table 1 vetsci-10-00463-t001:** Distance, expressed in centimeters, between the pins. Letters from a to i represent the distances between the pins described in Figure 2.

Distances between the Pins	Values (cm)
a	5
b	3
c	2
d	4
e	2
f	3
g	2
h	1.5
i	3

## Data Availability

The data presented in this study are available in this article.

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
