# Peer review of "Surgical Management of a Salter-Harris Type I Distal Physeal Fracture of the Tibia in a Calf: A Case Report"

_vetsci, 2023, doi:10.3390/vetsci10070463_

Round 1

Reviewer 1 Report

It is an original manuscript that will help to use this technique in the future for similar cases and hopefully be successful like this one. However, the manuscript requires major revision from the medical and English point of view. 

First of all, I believe the diagnosis is incorrect looking at the radiographs it seems to me that this calf suffered a complete physeal fracture through the distal physis of the tibia. Dislocation will be through the joint. And the description should be based on the distal fragment (as far as lateral to medial and dorsal to palmar). 

It is very confusing the description of the surgery technique used in this case report. The authors are talking about the use of wires and in the middle is talking about the pins. I highly recommend to re-write all this section and specifically to go step by step and use the appropriate nomenclature (please revise any small animal othropedic fracture repair book using this type of external fixation). 

The radiograph should be labeled lateromedial and dorsopalmar. 

I do recommend to re-structure the entire discussion section. 

The manuscript needs to be re-written due to major mistakes using English grammar. I do recommend to have the entire manuscript revise by some English speaking person before it is re-submitted.  

Author Response

It is an original manuscript that will help to use this technique in the future for similar cases and hopefully be successful like this one. However, the manuscript requires major revision from the medical and English point of view. 

AU: thank you for your comment. Authors consequently tried to significantly improve the level of English language.

First of all, I believe the diagnosis is incorrect looking at the radiographs it seems to me that this calf suffered a complete physeal fracture through the distal physis of the tibia. Dislocation will be through the joint. And the description should be based on the distal fragment (as far as lateral to medial and dorsal to palmar). 

AU: thank you for your very pertinent comment. Authors agree with your diagnosis and are very confused about this error. However, the technique used remains original and the authors are convinced that it may be useful for future similar cases. The present manuscript has therefore been revised to take account of the new diagnosis.

It is very confusing the description of the surgery technique used in this case report. The authors are talking about the use of wires and in the middle is talking about the pins. I highly recommend to re-write all this section and spsecifically to go step by step and use the appropriate nomenclature (please revise any small animal othropedic fracture repair book using this type of external fixation). 

AU: Authors tried to explain the surgical technique with more clarity.

The radiograph should be labeled lateromedial and dorsopalmar. 

AU: the incidences used for radiographs were latero-medial and cranio-caudal, respectively. They were adequately added to the manuscript

I do recommend to re-structure the entire discussion section.

AU: the discussion was entirely revised, according to the new diagnosis.

Reviewer 2 Report

Suggest the authors read my attempt at improving the paper by eliminating repetition, using more precise English , removing redundant words and phrases. They need to learn from my suggestions and rewrite parts of the paper as indicated.

This is an excellent case to report on, and a privilege to  review. Having said that, the paper itself suffers from poor use of English, much repetition, perhaps imprecise terminology. The following is my attempt to address these problems which you can accept or modify as the authors.

Tibiotalus dislocation in a calf treated and managed using external fixators: a case report.

Abstract

Generally, joint dislocation are not common in cattle: those involving the tibiotalus are, indeed, rare in other animal species and humans. A calf with a left tibiotalus dislocation was referred to…… Conservative manipulation and treatment failed to correct it and a novel technique was devised using external………stabilise it. Six weeks after treatment, the calf recovered successfully the use of the affected limb without any adverse sequelae. This innovative surgical technique may be used….. future when a simple closed reduction is impossible. 

Introduction

Lameness in cattle is a common clinical entity: with calves, the most common causes are associated with bone fractures and arthritis (2). To the best of the authors’ knowledge, dislocation of the tibiotalus in cattle has never been described , because of the comparative strength of the interosseous ligaments of the tarsus compared to bone, which is the same in human orthopaedics (3; 4; 5) where ligaments appear to be more difficult to break than bone. Two cases of inter tarsal dislocation in heifers have been reported but their successful treatments were non-surgical (10).

The present paper describes…… a tibiotalus dislocation in a pedigree heifer calf using external fixators.

Case presentation

….female Blonde d’Aquitaine calf’s referred to…….France, following sudden onset of lameness involving the left hind limb two days previously. The farmer had not observed any prior trauma to the heifer and throughout its illness had suckled its dam with vigour.

On arrival at the clinic, the calf weighed 80kg. A general clinical examination revealed no abnormal findings other than being unable to support the affected left hind limb: when asked to move, a severe lameness was  observed. However, its behaviour appeared relatively unaffected as it tried to walk, could stand, lie down and get up, all unaided. The left hock presented as swollen with a haematoma over the distal tibia: and there was an abnormal angle inwards towards the medial aspect of the limb.There was no…. splintering nor break in the skin over the joint. A provisional diagnosis included….., or either an inter tarsal or tibia-tarsal dislocation. Routine cranio-caudal and lateral radiographs were taken, and revealed a medic-caudal dislocation of the tibiotalus (see Figure 1)

Caption for Figure 1. Radiograph of the left hock joint of a 35 day-old heifer calf. A) cranio-caudal view with medial displacement of the tibia; b) lateral view with posterio-medial displacement, both relative to the talus.

Treatment

Prior to treatment, prophylactic benzylpenicillin dihydrostreptomycin (Intrmicine: CEVA) was administered routinely at a dose of 20mg/kg body weight b. wt) , followed by 0.5mg/kg b. wt meloxicam (Recocam: Bimeda). General anaesthesia was induced using 0.025mg/kg b.wt  xylazine (Sedaxylan: Dechra) intravenously, 0.06mg/kg b. wt butorphanol (Torbugesic: Zoetis) and 2.9mg/kg b.wt ketamine (Ketamine 1000: Virbac) (10): a 12mm diameter endo-tracheal tube was placed routinely and anaesthesia maintained using isoflurane (IsoFlo: Zoetis) and oxygen. To further relax the left hind limb, procaine (Procamidor: Axience) was administered in the epidural space (say where - lumbosacral or higher and what volume?). Throughout the procedure, 0.9% isotonic saline was administered through an intravenous drip at a rate of 10ml/kg/hr.

 First, a simple manual reduction was attempted but this was unsuccessful, due to the degree of tibial displacement and the severe and persistent contraction of  the gastrocnemeus muscle and others of the upper thigh.  With the owner’s consent, a surgical approach to devised to treat and manage this dislocation.

The entire left hind limb from the stifle to the hoof was prepared routinely for surgery, the calf placed in right  latero-dorsal recumbency. Six-millimetre stainless steel and titanium pins were placed  into the lateral aspect of the distal tibia, the proximal metatarsal, and the calcaneus according to Figure 2 and Table 1. (No numbers in the Figure: label a,b, c  etc, and use table to state distances between  each pin, and between distal tibial pin and lateral malleleus, and proximal metatarsal pin and central tarsal.  Great figure 2. Caption. Figure 2. Schematic lateral view of pin locations for the supporting external scaffold to attach to: after Barone (12). The angle at which the pins were inserted  must be clarified so that the external scaffold  was firm. This means that Figure 3 the assembly using radiographs could be referred to earlier in the text.

 Initially, attempts to reduce the dislocation were made using leverage on the pins themselves, but this was unsuccessful. A 7cm-long incision was made medially …now I am lost because a diagram may better describe where incisions were made, I have never encountered Hohmannn or his levers but this whole paragraph needs clarity, nothing in brackets.

 Caption Figure 3. The radiographs should be a) and b) and the gross limb photographs c) and d). Is there a medial photograph? I think it is important to indicate the angles of the scaffold supporting the joint in semi-flexion and included in table !. It is all about organising the text and using the Figures to illustrate what went on!

  1. Cranio-caudal and b) lateral radiographs describing the pins and external scaffold supporting the repositioned tarsotalus joint; c) and d) showing the actual pins and scaffold in place and the angles of the scaffold relative to the  tarsi-metatarsal joint.

I do not understand the bandage and mount, crepe etc. Do not detail the disinfection process it is all routine surgical procedure.

Post-operative treatment included  intravenous 3000IU/kg b.wt gentamicin (G4; Virbac) for three days after surgery, and continuance of benxylpenicillin dihydrostreptomycin for 20 days. The superficial dressings were changed routinely on days two, seven and nineteen after surgery. Recovery was uneventful, confirmed by radiographs taken (state when)

 Six weeks after surgery, the heifer was sedated using xylazine as before, the scaffold was dismantled and pins removed, and radiographs taken (see Figure 4)

Caption. Figure 4: a) lateral and b) cranio-caudal radiographs showing conformation of the tibio-tarsal-metatarsal joint following removal of the pins and scaffold and the absence of significant periostitis or lysis of bone cortex.  

The heifer was returned home 69 days following treatment. Atrophy of the muscles of the left pelvic limb was discernible, and she walked with a persistent limp: she was closely confined in a well-bedded loose box and physiotherapy was initiated on the farm. Four months after surgery, the physical appearance of the heifer appeared normal and the limp had disappeared (Figure 5).

Caption. Figure 5: five month-old Blonde d’Aquitaine heifer following surgical treatment of a left tibiotalus dislocation. There remains slight left hind limb extension at rest. 

Discussion

Needs rewriting in a more direct style of English.

Para 1: the most important outcome of this case - successful treatment. How achieved, tricks used by surgeons, previous orthopaedic experience of external fixation of fractures dislocations in equines? Large dogs? Do not digress.

Para 2: second most important feature - post-operative management. Hygiene, close confinement, physiotherapy

Para 3: describe in specific terms the economic argument for the treatment. Essential if you want others to try the external fixation technique. Genetic value, future breeding schedule.

Para 4: compare this treatment with that in other species - dog, human. Why this calf posed difficulties - weight borne on limb, muscle contraction prior to surgery due to time lag and forces acting on the joint affecting repositioning of dislocation, not an issue with dogs or humans because nursing post-operatively is easy!!

Para 5: how could this outcome be achieved outside a university veterinary hospital with lots of professional skilled staff. If not in practice conditions, what must a practitioner do to ensure best outcome. All about early decision making.

References

Just check to ensure each is relevant to the precise point you are making. We all know that cattle lameness is common and you don’t need a prevalence paper on cattle lameness to support that statement! Please ensure each reference is needed to support a point. To me references 12, 16, 24, 27, and 28 seem irrelevant and add very little to the paper.

I hope this helps.

Author Response

Suggest the authors read my attempt at improving the paper by eliminating repetition, using more precise English, removing redundant words and phrases. They need to learn from my suggestions and rewrite parts of the paper as indicated.

AU: thank you for your comment. Authors tried to significantly improve the level of English, notably by integrating your excellent suggestions. Thank you for your help in this revision.

This is an excellent case to report on, and a privilege to  review. Having said that, the paper itself suffers from poor use of English, much repetition, perhaps imprecise terminology. The following is my attempt to address these problems which you can accept or modify as the authors.

Tibiotalus dislocation in a calf treated and managed using external fixators: a case report.

AU: thank you for your suggestion. After revision, authors realized that the initial diagnosis was wrong and are very sorry for this error. The diagnosis is a Salter-Harris type I physeal fracture of the distal tibia. However, the technique remains innovative and may be useful for veterinary practitioners in future similar cases. The present manuscript has been carefully revised, taking into account as far as possible all your pertinent comments.

Abstract

Generally, joint dislocation are not common in cattle: those involving the tibiotalus are, indeed, rare in other animal species and humans. A calf with a left tibiotalus dislocation was referred to…… Conservative manipulation and treatment failed to correct it and a novel technique was devised using external………stabilise it. Six weeks after treatment, the calf recovered successfully the use of the affected limb without any adverse sequelae. This innovative surgical technique may be used….. future when a simple closed reduction is impossible. 

AU: your suggestions were added in the manuscript.

Introduction

Lameness in cattle is a common clinical entity: with calves, the most common causes are associated with bone fractures and arthritis (2). To the best of the authors’ knowledge, dislocation of the tibiotalus in cattle has never been described , because of the comparative strength of the interosseous ligaments of the tarsus compared to bone, which is the same in human orthopaedics (3; 4; 5) where ligaments appear to be more difficult to break than bone. Two cases of inter tarsal dislocation in heifers have been reported but their successful treatments were non-surgical (10).

The present paper describes…… a tibiotalus dislocation in a pedigree heifer calf using external fixators.

AU: your suggestions were added in the manuscript.

Case presentation

….female Blonde d’Aquitaine calf’s referred to…….France, following sudden onset of lameness involving the left hind limb two days previously. The farmer had not observed any prior trauma to the heifer and throughout its illness had suckled its dam with vigour.

On arrival at the clinic, the calf weighed 80kg. A general clinical examination revealed no abnormal findings other than being unable to support the affected left hind limb: when asked to move, a severe lameness was  observed. However, its behaviour appeared relatively unaffected as it tried to walk, could stand, lie down and get up, all unaided. The left hock presented as swollen with a haematoma over the distal tibia: and there was an abnormal angle inwards towards the medial aspect of the limb.There was no…. splintering nor break in the skin over the joint. A provisional diagnosis included….., or either an inter tarsal or tibia-tarsal dislocation. Routine cranio-caudal and lateral radiographs were taken, and revealed a medic-caudal dislocation of the tibiotalus (see Figure 1)

AU: your suggestions were added in the manuscript.

Caption for Figure 1. Radiograph of the left hock joint of a 35 day-old heifer calf. A) cranio-caudal view with medial displacement of the tibia; b) lateral view with posterio-medial displacement, both relative to the talus.

AU: your suggestions were added in the manuscript.

Treatment

Prior to treatment, prophylactic benzylpenicillin dihydrostreptomycin (Intrmicine: CEVA) was administered routinely at a dose of 20mg/kg body weight b. wt) , followed by 0.5mg/kg b. wt meloxicam (Recocam: Bimeda). General anaesthesia was induced using 0.025mg/kg b.wt  xylazine (Sedaxylan: Dechra) intravenously, 0.06mg/kg b. wt butorphanol (Torbugesic: Zoetis) and 2.9mg/kg b.wt ketamine (Ketamine 1000: Virbac) (10): a 12mm diameter endo-tracheal tube was placed routinely and anaesthesia maintained using isoflurane (IsoFlo: Zoetis) and oxygen. To further relax the left hind limb, procaine (Procamidor: Axience) was administered in the epidural space (say where - lumbosacral or higher and what volume?). Throughout the procedure, 0.9% isotonic saline was administered through an intravenous drip at a rate of 10ml/kg/hr.

First, a simple manual reduction was attempted but this was unsuccessful, due to the degree of tibial displacement and the severe and persistent contraction of  the gastrocnemeus muscle and others of the upper thigh.  With the owner’s consent, a surgical approach to devised to treat and manage this dislocation.

The entire left hind limb from the stifle to the hoof was prepared routinely for surgery, the calf placed in right  latero-dorsal recumbency. Six-millimetre stainless steel and titanium pins were placed  into the lateral aspect of the distal tibia, the proximal metatarsal, and the calcaneus according to Figure 2 and Table 1. 

(No numbers in the Figure: label a,b, c  etc, and use table to state distances between  each pin, and between distal tibial pin and lateral malleleus, and proximal metatarsal pin and central tarsal. 

Great figure 2. Caption. Figure 2. Schematic lateral view of pin locations for the supporting external scaffold to attach to: after Barone (12). The angle at which the pins were inserted  must be clarified so that the external scaffold  was firm. This means that Figure 3 the assembly using radiographs could be referred to earlier in the text.

AU: your suggestions were added in the manuscript. Regarding the angles between the different pins, they were unfortunately not measured on the day of surgery. However, strict respect to these angles in future similar cases is not very crucial, given that each situation is unique and requires some adaptations by surgeons. The crucial point is to respect the non-parallel application of the pins in order to create a triangle. This configuration is essential by generating a solid overall assembly. Some sentences about that were accordingly added in the discussion.

Initially, attempts to reduce the dislocation were made using leverage on the pins themselves, but this was unsuccessful. A 7cm-long incision was made medially …now I am lost because a diagram may better describe where incisions were made, I have never encountered Hohmannn or his levers but this whole paragraph needs clarity, nothing in brackets.

AU: your suggestions were added in the manuscript. Authors also tried to clarify the description.

Caption Figure 3. The radiographs should be a) and b) and the gross limb photographs c) and d). Is there a medial photograph? I think it is important to indicate the angles of the scaffold supporting the joint in semi-flexion and included in table !. It is all about organising the text and using the Figures to illustrate what went on!

  1. Cranio-caudal and b) lateral radiographs describing the pins and external scaffold supporting the repositioned tarsotalus joint; c) and d) showing the actual pins and scaffold in place and the angles of the scaffold relative to the  tarsi-metatarsal joint.

AU: your comments and suggestions were added in the manuscript.

I do not understand the bandage and mount, crepe etc. Do not detail the disinfection process it is all routine surgical procedure.

Post-operative treatment included  intravenous 3000IU/kg b.wt gentamicin (G4; Virbac) for three days after surgery, and continuance of benxylpenicillin dihydrostreptomycin for 20 days. The superficial dressings were changed routinely on days two, seven and nineteen after surgery. Recovery was uneventful, confirmed by radiographs taken (state when)

 Six weeks after surgery, the heifer was sedated using xylazine as before, the scaffold was dismantled and pins removed, and radiographs taken (see Figure 4)

Caption. Figure 4: a) lateral and b) cranio-caudal radiographs showing conformation of the tibio-tarsal-metatarsal joint following removal of the pins and scaffold and the absence of significant periostitis or lysis of bone cortex.  

The heifer was returned home 69 days following treatment. Atrophy of the muscles of the left pelvic limb was discernible, and she walked with a persistent limp: she was closely confined in a well-bedded loose box and physiotherapy was initiated on the farm. Four months after surgery, the physical appearance of the heifer appeared normal and the limp had disappeared (Figure 5).

Caption. Figure 5: five month-old Blonde d’Aquitaine heifer following surgical treatment of a left tibiotalus dislocation. There remains slight left hind limb extension at rest. 

AU: all your comments and suggestions were included in the manuscript. Authors warmly thank you for your help.

Discussion

Needs rewriting in a more direct style of English.

Para 1: the most important outcome of this case - successful treatment. How achieved, tricks used by surgeons, previous orthopaedic experience of external fixation of fractures dislocations in equines? Large dogs? Do not digress.

Para 2: second most important feature - post-operative management. Hygiene, close confinement, physiotherapy

Para 3: describe in specific terms the economic argument for the treatment. Essential if you want others to try the external fixation technique. Genetic value, future breeding schedule.

Para 4: compare this treatment with that in other species - dog, human. Why this calf posed difficulties - weight borne on limb, muscle contraction prior to surgery due to time lag and forces acting on the joint affecting repositioning of dislocation, not an issue with dogs or humans because nursing post-operatively is easy!!

Para 5: how could this outcome be achieved outside a university veterinary hospital with lots of professional skilled staff. If not in practice conditions, what must a practitioner do to ensure best outcome. All about early decision making.

AU: all the aspects that you cited above for the discussion were including in the manuscript.

References

Just check to ensure each is relevant to the precise point you are making. We all know that cattle lameness is common and you don’t need a prevalence paper on cattle lameness to support that statement! Please ensure each reference is needed to support a point. To me references 12, 16, 24, 27, and 28 seem irrelevant and add very little to the paper.

AU: thank you for your comment. Some of them were accordingly removed. For the other ones, they were kept in the manuscript for two reasons: 1/ only a few specific data on this general approach (including both surgery and medications used) are available on cattle, making it necessary to use references in other species. 2/ the supportive references used justify the general approach adopted in this case.

I hope this helps.

AU: your consistent comments and suggestions were very helpful. Thank you very much!

Round 2

Reviewer 1 Report

The authors have done significant modifications and looks correct to me. 

This has been improved compared to the previous version and it is now ready for submission

Author Response

The authors have done significant modifications and looks correct to me. 

This has been improved compared to the previous version and it is now ready for submission.

AU: thank you very much for helping us in the preparation of the present manuscript.

Reviewer 2 Report

Introduction:

ll 31- 32  ……can vary widely…..type, and nature: those associated with the tibia are, perhaps, most frequently encountered. (For the future: adverbs usually follow the verb they qualify. “Fractures” are written three times in two lines!)

ll 33.   Are you sure about this statement? Epiphyseal fracture of long bones generally are common in calves/lambs/puppies/children following external forces applied to long bones.

ll 34   ….described in…..

ll 36 - 37 ….a very small number of similar fractures have been described which suggests the case reported in this paper is rare.

ll 44 ….limb two days…. (For all small numbers and those at the very beginning of a sentence, write the number in letters , not as a figure)

ll 51….: there was….of the hock inwards….  (A colon : means you do not have to place the conjunction and  following it. The two phrases either side of a colon refer to the same subject at the start of the sentence. It makes for an easier read!)

ll 55.  ….itself, or either….

ll 56 - 58. A definitive diagnosis was made following routine …..radiographs that revealed….

ll 60 - 61. This sentence should be removed from the text and placed in the text for Figure 1 at the end. Please, you must not keep repeating Source: Clinic of Ruminants. Readers of this paper know who you all are and where the paper has originated from. It is rather irritating, so keep the critical reader on your side: they will respect you for it!

ll 67 - 71.  The severe medial displacement of the tarsal-tibio tarsal joint and the inability to reposition it by traction ruled out the possibility of treating this fracture conservatively, hence a surgical approach was deemed necessary to restore permanent good function to the limb of this calf - having a high genetic value - with the agreement of its owner. 

ll 72 remove ‘association’: just name the drugs.

ll 82. ….surgery, and the……

ll 85   …calcaneus (se Figure 2 and Table 1). 

ll 86 - 89.  This entire paragraph should go to ll 81, where you explain the objective of the surgery before you start1 The objective of the surgical  procedure was to use…Explain JAM.

From this point on, all the text MUST be in the past tense - ‘was’ NOT ‘is’. It now becomes confused when we have titanium pins and Kirschner pins: aren’t they the same? I don’t understand ll 98/99: what is the penetration?

ll 117 The pins acted: first, as a lever point to reduce the fracture; second, as fixators for the external scaffold to immobilise the joint.

ll 120. ….made precise reduction of the fracture very difficult. (Delicate suggests gentleness, delicacy: this needed considerable sustained force and precision)

ll 124 - 126 This is unclear. You cannot perform a lever, you can insert a lever and then do something with it. I sympathise with you: finding precise English words is difficult and I could not write anything in French!!! But you have to try and I am trying to help. If you wrote in French, I would have no idea of the exact meaning you are trying to get over. Would a human orthopaedic textbook in English  help, particularly when you are describing a named specific procedure?

ll 132 onwards.  Once the tibial-tibia tarsal joint had been replaced in its anatomically correct position, the 6-mm union bars WERE…..All this must be in the past tense

ll 140  Post-operative routine radiological examination confirmed that this joint had been repositioned correctly (see Figure 3 a & b).

ll 141 - 143. This sentence should go in the discussion, where you reflect on the surgical difficulties that were encountered. 

ll 152 Finally, the skin over the entire hock joint was bandaged below  the scaffold??? To provide a clean….

ll 170 Remove Source: etc You’ve said it before!

My effort.

This paper describes the successful management of a complete physeal fracture regarding the distal physis of the tibia. In cattle, the most frequent cause of such fractures is excessive and misdirected traction applied to overcome fetal posterior presentation and relative oversize in dystocia cases |16-18]. In comparison, tibial physeal fractures occur in around 10% of cases in foals [19], most frequently as a result of external traumatic events [20]. In addition, such severe displacement of both bone segments 192 on either side of the fracture has not been described before, making the present case 193 report unique. (Contradicts your Introduction!!)

The se- 202 vere displacement of the two bone segments on either side of the fracture, combined with 203 the need for a solid assembly to take account of the vertical and horizontal forces applied to the pelvic limb 204 in cattle as they walk, justified the complex assembly described in this paper. The three most 206 important elements in this assembly, which guarantee its solidity, are a) the location of the 207 pins, b) the distance between all of them and c) the configuration of the triangle on either 208 side of the fracture for the scaffolding to be secure. The value of the angles between each pin is not crucial, as it can vary 209 from one animal to another, depending on its conformation and the specific characteriza- 210 tion of the fracture. 211 

The success of the treatment depends also on the management of the post- 213 operative period. This calf was kept in a clean box with a severe restriction to its movements that encouraged healing. Both frequent new bandage applications and physiother- 215 apy after pin removal contributed to the positive outcome of this case. Physeal fractures 216 usually heal within four weeks [26], whereas the  recovery period in this case lasted six week:. in comparison, foals recover from similar conditions within 6 to 8 weeks [27]. 218 As the displacement of the two bone segments on either side of the fracture was severe, an 219 additional two-week extension was given to ensure proper fracture healing before removal of 220 the external scaffold. 

Post-operative complications must be considered when discussing the prognosis in such cases. In all species, the major compli- 222 cation is joint stiffness, associated with lameness in some cases, especially when pins are 223 applied more than four weeks [28]. Full recovery is usually quite good after several months. 224 Paraesthesia of the tibial or dorsal cutaneous nerve may be encountered in around 25% of open dislocation found in humans together with arthritis and degenerative osteoarthritis, in 25% of cases of open dislocations [29]. In this case, the 226 use of fixators could have led to osteomyelitis but maintaining optimum sterility and cleanliness both during surgery and post-operatively ensured a successful treatment outcome. Antibiotic 228 prophylaxis is recommended for all surgery involving the placement of implants [30]. 229 First-generation cephalosporins, cefazoline and cefalexine are the antibiotics of choice [31,32]. But their use is 231 prohibited in cattle (in which countries???) , so there are no maximum residue limit values for these two molecules. 232 Whittem et al. (1999) [33] showed no difference in efficacy between the use of penicillin 233 and cephalosporin in orthopedic surgery in dogs. The combination of penicillin and gen- 234 tamicin broadened the spectrum of action. Gentamicin was used for only 3 days due 235 to risk of nephrotoxicity [34]. 236 

pastedGraphic.png

The cost-effectiveness of such an intervention must be considered. This treatment was 237 agreed with the farmer who accepted the financial cost that was sub- 238 stantial. Adapta- 239 tions are possible to reduce the price. For example, the animal remained in hospital 240 throughout its convalescence but returning home was placed in a calving pen. 241 This option should be favored in field conditions, by educating the farmer about post-op- 242 erative procedures associated and regular checks by the veterinarian during the recovery 243 period. In the present situation, the positive outcome and the high genetic value of the 244 animal made the care clearly profitable.

What is missing - specific actions that were taken to determine where the pins should be placed either side of the fracture site, and the method of their insertion; the genetic value of the heifer, for the herd and productivity or the breeding value for selling on progeny? Also, antibiotics to reduce secondary infection post-operatively - their concentrations in wounds and bone - their ability to eliminate Gram +ve and Gram -ve infections: can they be used in all countries where the expertise to carry out this type of surgery is found?.  

See above 

Author Response

Introduction:

ll 31- 32  ……can vary widely…..type, and nature: those associated with the tibia are, perhaps, most frequently encountered. (For the future: adverbs usually follow the verb they qualify. “Fractures” are written three times in two lines!)

ll 33.   Are you sure about this statement? Epiphyseal fracture of long bones generally are common in calves/lambs/puppies/children following external forces applied to long bones.

ll 34   ….described in…..

ll 36 - 37 ….a very small number of similar fractures have been described which suggests the case reported in this paper is rare.

ll 44 ….limb two days…. (For all small numbers and those at the very beginning of a sentence, write the number in letters , not as a figure)

ll 51….: there was….of the hock inwards….  (A colon : means you do not have to place the conjunction and  following it. The two phrases either side of a colon refer to the same subject at the start of the sentence. It makes for an easier read!)

ll 55.  ….itself, or either….

ll 56 - 58. A definitive diagnosis was made following routine …..radiographs that revealed….

ll 60 - 61. This sentence should be removed from the text and placed in the text for Figure 1 at the end. Please, you must not keep repeating Source: Clinic of Ruminants. Readers of this paper know who you all are and where the paper has originated from. It is rather irritating, so keep the critical reader on your side: they will respect you for it!

ll 67 - 71.  The severe medial displacement of the tarsal-tibio tarsal joint and the inability to reposition it by traction ruled out the possibility of treating this fracture conservatively, hence a surgical approach was deemed necessary to restore permanent good function to the limb of this calf - having a high genetic value - with the agreement of its owner. 

ll 72 remove ‘association’: just name the drugs.

ll 82. ….surgery, and the……

ll 85   …calcaneus (se Figure 2 and Table 1). 

ll 86 - 89.  This entire paragraph should go to ll 81, where you explain the objective of the surgery before you start1 The objective of the surgical  procedure was to use…Explain JAM.

From this point on, all the text MUST be in the past tense - ‘was’ NOT ‘is’. It now becomes confused when we have titanium pins and Kirschner pins: aren’t they the same? I don’t understand ll 98/99: what is the penetration?

ll 117 The pins acted: first, as a lever point to reduce the fracture; second, as fixators for the external scaffold to immobilise the joint.

ll 120. ….made precise reduction of the fracture very difficult. (Delicate suggests gentleness, delicacy: this needed considerable sustained force and precision)

ll 124 - 126 This is unclear. You cannot perform a lever, you can insert a lever and then do something with it. I sympathise with you: finding precise English words is difficult and I could not write anything in French!!! But you have to try and I am trying to help. If you wrote in French, I would have no idea of the exact meaning you are trying to get over. Would a human orthopaedic textbook in English  help, particularly when you are describing a named specific procedure?

ll 132 onwards.  Once the tibial-tibia tarsal joint had been replaced in its anatomically correct position, the 6-mm union bars WERE…..All this must be in the past tense

ll 140  Post-operative routine radiological examination confirmed that this joint had been repositioned correctly (see Figure 3 a & b).

ll 141 - 143. This sentence should go in the discussion, where you reflect on the surgical difficulties that were encountered. 

ll 152 Finally, the skin over the entire hock joint was bandaged below  the scaffold??? To provide a clean….

ll 170 Remove Source: etc You’ve said it before!

My effort.

This paper describes the successful management of a complete physeal fracture regarding the distal physis of the tibia. In cattle, the most frequent cause of such fractures is excessive and misdirected traction applied to overcome fetal posterior presentation and relative oversize in dystocia cases |16-18]. In comparison, tibial physeal fractures occur in around 10% of cases in foals [19], most frequently as a result of external traumatic events [20]. In addition, such severe displacement of both bone segments on either side of the fracture has not been described before, making the present case report unique.

The severe displacement of the two bone segments on either side of the fracture, combined with the need for a solid assembly to take account of the vertical and horizontal forces applied to the pelvic limb in cattle as they walk, justified the complex assembly described in this paper. The three most important elements in this assembly, which guarantee its solidity, are a) the location of the pins, b) the distance between all of them and c) the configuration of the triangle on either side of the fracture for the scaffolding to be secure. The value of the angles between each pin is not crucial, as it can vary from one animal to another, depending on its conformation and the specific characterization of the fracture.

The success of the treatment depends also on the management of the post-operative period. This calf was kept in a clean box with a severe restriction to its movements that encouraged healing. Both frequent new bandage applications and physiotherapy after pin removal contributed to the positive outcome of this case. Physeal fractures usually heal within four weeks [26], whereas the recovery period in this case lasted six weeks: in comparison, foals recover from similar conditions within 6 to 8 weeks [27]. As the displacement of the two bone segments on either side of the fracture was severe, an additional two-week extension was given to ensure proper fracture healing before removal of the external scaffold. 

Post-operative complications must be considered when discussing the prognosis in such cases. In all species, the major complication is joint stiffness, associated with lameness in some cases, especially when pins are applied more than four weeks [28]. Full recovery is usually quite good after several months. Paraesthesia of the tibial or dorsal cutaneous nerve may be encountered in around 25% of open dislocation found in humans together with arthritis and degenerative osteoarthritis, in 25% of cases of open dislocations [29]. In this case, the use of fixators could have led to osteomyelitis but maintaining optimum sterility and cleanliness both during surgery and post-operatively ensured a successful treatment outcome. Antibiotic prophylaxis is recommended for all surgery involving the placement of implants [30]. First-generation cephalosporins, cefazoline and cefalexine are the antibiotics of choice [31,32]. But their use is prohibited in cattle (in which countries???), so there are no maximum residue limit values for these two molecules. Whittem et al. (1999) [33] showed no difference in efficacy between the use of penicillin and cephalosporin in orthopedic surgery in dogs. The combination of penicillin and gentamicin broadened the spectrum of action. Gentamicin was used for only 3 days due to risk of nephrotoxicity [34].

The cost-effectiveness of such an intervention must be considered. This treatment was agreed with the farmer who accepted the financial cost that was substantial. Adaptations are possible to reduce the price. For example, the animal remained in hospital throughout its convalescence but returning home was placed in a calving pen. This option should be favored in field conditions, by educating the farmer about post-operative procedures associated and regular checks by the veterinarian during the recovery period. In the present situation, the positive outcome and the high genetic value of the animal made the care clearly profitable.

What is missing - specific actions that were taken to determine where the pins should be placed either side of the fracture site, and the method of their insertion;

the genetic value of the heifer, for the herd and productivity or the breeding value for selling on progeny?

Also, antibiotics to reduce secondary infection post-operatively - their concentrations in wounds and bone - their ability to eliminate Gram +ve and Gram -ve infections: can they be used in all countries where the expertise to carry out this type of surgery is found?

AU: all the elements you suggest in your review were included in the manuscript, with supportive references if necessary. We hope that the paper gained in clarity after these changes. The authors would like to thank you for your invaluable help in improving this manuscript, both in the English language and on substantial elements.